# Study protocol of the ASTOP trial: A multicenter, randomized, double-blind, placebo-controlled trial of presurgical aspirin administration for the prevention of thromboembolic complications of coil embolization for ruptured aneurysms

Sakyo Hirai[1], Kyohei Fujita[1], Shoko Fujii[1], Satoru Takahashi[1], Keigo Shigeta[2], Jun Karakama[3], Yukiko Enomoto[4], Yohei Sato[5], Masataka Yoshimura[6,7], Shin Hirota[6], Tatsuya Mizoue[8], Yoshikazu Yoshino[9], Yoshihisa Kawano[10,11], Toshihiro Yamamura[10], Shinya Kohyama[7], Masaru Hirohata[12], Shinichi Yoshimura[13], Yosuke Ishii[14], Toshihiro Yamauchi[15], Naoki Taira[16], Yoshiki Obata[17], Makoto Sakamoto[18], Masato Inoue[19], Motoshige Yamashina[20], So Tokunaga[21], Toshio Higashi[22], Kana Sawada[23], Hidetoshi Mochida[24], Keisuke Ido[25], Masataka Takeuchi[26], Tomoji Takigawa[27], Yasushi Takagi[28], Masafumi Morimoto[29], Masataka Nanto[30], Kazunori Miki[31], Kouichi Misaki[32], Koichi Arimura[33], Yoshiki Hanaoka[34], Mutsuya Hara[35], Shoko Hara[11], Kota Yokoyama[36], Jun Ooyama[36], Ryoichi Hanazawa[37], Hiroyuki Sato[37], Akihiro Hirakawa[37], Megumi Ishiguro[38], Shigeru Nemoto[14], Kazutaka Sumita[1]*

1 Department of Endovascular Surgery, Tokyo Medical and Dental University, Tokyo, Japan, 2 Department of Neurosurgery, National Hospital Organization Disaster Medical Center, Tokyo, Japan, 3 Department of Neurosurgery, Ome Medical Center, Tokyo, Japan, 4 Department of Neurosurgery, Gifu University Graduate School of Medicine, Gifu, Japan, 5 Department of Neurosurgery, Japanese Red Cross Musashino Hospital, Tokyo, Japan, 6 Department of Neurosurgery, Tsuchiura Kyodo General Hospital, Ibaraki, Japan, 7 Department of Endovascular Neurosurgery, Saitama Medical University International Medical Center, Saitama, Japan, 8 Department of Neurosurgery, Suiseikai Kajikawa Hospital, Hiroshima, Japan, 9 Department of Neuroendovascular Surgery, Jichi Medical University Saitama Medical Center, Saitama, Japan, 10 Department of Neurosurgery, JA Toride Medical Center, Ibaraki, Japan, 11 Department of Neurosurgery, Tokyo Medical and Dental University, Tokyo, Japan, 12 Department of Neurosurgery, Kurume University Hospital, Fukuoka, Japan, 13 Department of Neurosurgery, Hyogo Medical University Hospital, Hyogo, Japan, 14 Department of Neurosurgery, Kanto Rosai Hospital, Kanagawa, Japan, 15 Department of Neurosurgery, Chiba Emergency and Psychiatric Medical Center, Chiba, Japan, 16 Department of Neurosurgery, Shuwa General Hospital, Saitama, Japan, 17 Department of Neurosurgery, Tokyo Kita Medical Center, Tokyo, Japan, 18 Faculty of Medicine, Department of Brain and Neurosciences, Division of Neurosurgery, Tottori University, Tottori, Japan, 19 Department of Neurosurgery, Center Hospital of the National Center for Global Health and Medicine, Tokyo, Japan, 20 Department of Neurosurgery, Soka Municipal Hospital, Saitama, Japan, 21 Department of Neuroendovascular Surgery, National Hospital Organization Kyushu Medical Center, Fukuoka, Japan, 22 Department of Neurosurgery, Fukuoka University Chikushi Hospital, Fukuoka, Japan, 23 Department of Neurosurgery, Tokyo Bay Urayasu Ichikawa Medical Center, Chiba, Japan, 24 Department of Neurosurgery, Asahi General Hospital, Chiba, Japan, 25 Department of Neurosurgery, Saga Prefectural Hospital Koseikan, Saga, Japan, 26 Department of Neurosurgery, Seisho Hospital, Kanagawa, Japan, 27 Department of Neurosurgery, Dokkyo Medical University Saitama Medical Center, Saitama, Japan, 28 Department of Neurosurgery, Tokushima University Hospital, Tokushima, Japan, 29 Department of Neurosurgery, Yokohama Shintoshi Neurosurgical Hospital, Kanagawa, Japan, 30 Department of Neurosurgery, Graduate School of Medical Science, Kyoto Prefectural University of Medicine, Kyoto, Japan, 31 Department of Neuroendovascular Surgery, Tokyo Metropolitan Police Hospital, Tokyo, Japan, 32 Department of Neurosurgery, Kanazawa University Hospital, Kanazawa, Japan, 33 Department of Neurosurgery, Graduate School of Medical Sciences, Kyushu University, Fukuoka, Japan, 34 Department of Neurosurgery, Shinshu University Hospital, Nagano, Japan, 35 Department of Neurosurgery, Tokyo Metropolitan Toshima Hospital, Tokyo, Japan, 36 Department of Radiology, Tokyo



**Data Availability Statement:** No datasets were generated or analysed because this is a protocol paper for an ongoing randomized controlled trial.All relevant data from this study will be made available upon study completion.

**Funding:** The ASTOP is funded by the following financial support: the Japanese Society of Neuroendovascular Therapy and crowdfunding for ASTOP. The funders had no role in study design, data collection and analysis, decision to publish, or preparation of the manuscript. The ASTOP study was registered with the Japan Registry of Clinical Trials (identifier: jRCTs031210421).

**Competing interests:** The authors have declared that no competing interests exist.

Medical and Dental University, Tokyo, Japan, **37** Department of Clinical Biostatistics, Graduate School of Medical and Dental Sciences, Tokyo Medical and Dental University, Tokyo, Japan, **38** Department of Health Science and Development Center, Tokyo Medical and Dental University, Tokyo, Japan

* sumita.nsrg@tmd.ac.jp

# Abstract

## Rationale

Thromboembolism is a serious complication of endovascular treatment for ruptured cerebral aneurysms. The administration of antiplatelet agents before endovascular treatment for ruptured cerebral aneurysms may reduce the risk of thromboembolic complications.

## Aim

This study aimed to assess the safety and efficacy of preoperative aspirin administration in endovascular treatment for ruptured cerebral aneurysms.

## Sample size estimates

Assuming a 15% incidence rate of both intraoperative thromboembolic morbidity and symptomatic ischemic lesions on magnetic resonance imaging diffusion-weighted imaging scans assessed by an Independent Review Committee, a sample size of 484 will be required to detect a 10% improvement with aspirin administration with 90% power using the Pearson's chi-square test at a two-sided significance level of 2.5% for each primary outcome, after accounting for a 5% dropout rate.

## Methods and design

ASTOP is a multicenter, randomized, double-blind, placebo-controlled clinical trial. A total of 484 patients with ruptured cerebral aneurysms receiving coil embolization within 72 h of onset will be randomly assigned 1:1 to receive 200 mg of aspirin or placebo before the procedure.

## Study outcomes

The primary outcomes will be the incidence rates of intraoperative thromboembolic complications and symptomatic ischemic lesions on magnetic resonance imaging diffusion-weighted imaging scans evaluated by the Independent Review Committee. The secondary outcomes will be the incidence rate of cerebral ischemic events and all bleeding events within 14 days of enrollment and functional outcomes defined by the modified Rankin Scale score at 90 days.

## Discussion

This trial will provide valuable data on the role of antiplatelet agents during endovascular treatment for ruptured cerebral aneurysms.

## Trial registration

**Registration**: Japan Registry of Clinical Trials, Identifier: jRCTs031210421.

# Introduction

Endovascular treatment (EVT) has become an established treatment for cerebral aneurysms [1]. Dual antiplatelet therapy with agents such as aspirin and clopidogrel is commonly used to prevent thromboembolic complications [2].

In the acute phase of ruptured aneurysms (RAs), EVT is associated with an increased incidence of intraoperative thromboembolic events due to the hypercoagulable state as a bioprotective response to subarachnoid hemorrhage (SAH) [3,4]. The administration of antiplatelet agents during EVT for RAs reduces thromboembolic complications [5]. In contrast, a subanalysis of the International Subarachnoid Aneurysm Trial reported that intraoperative or postoperative antiplatelet use of EVT for RAs did not improve clinical outcomes in patients with SAH [6]. However, a major limitation of that study was the timing of antiplatelet agent administration: antiplatelet agents should be preoperatively administered to prevent thrombus formation during the procedure. The potential for increased bleeding complications, including re-rupture of the aneurysm and gastrointestinal bleeding due to increased intracranial pressure, is a concern associated with the preoperative administration of antiplatelet agents that may promote bleeding in the acute phase of RAs. However, in previous retrospective studies on preoperative and intraoperative antiplatelet administration in EVT for RAs, hemorrhagic morbidity did not increase, and antiplatelet treatment was reported to be safe [5,7]. Nevertheless, this conclusion has not been verified in randomized controlled trials (RCTs).

Therefore, we designed this RCT to assess the safety and efficacy of preoperative antiplatelet therapy in patients who received EVT for RAs.

# Materials and methods

The ASTOP study (Japan Registry of Clinical Trials, Identifier: jRCTs031210421) is a multicenter, randomized, double-blind, placebo-controlled clinical trial. The trial is currently being conducted at 44 institutions in Japan, all of which have one or more specialists certified by the Japanese Society of Neuroendovascular Therapy. The study design was approved by the certified review board of Tokyo Medical and Dental University (approved number: NR2021-006) on September 17, 2021.

This study is conducted in accordance with the "Declaration of Helsinki" and "Ethical Guidelines for Clinical Research." Written informed consent is obtained from all patients before enrollment.

## Patient population

This study will include 484 patients (two treatment groups with a 1:1 randomization ratio) with RAs receiving EVT within 72 h of onset and receiving preoperative administration of 200 mg aspirin. Fig 1 presents a flowchart of the study.

The inclusion criteria are as follows:

1. RAs in the acute phase

2. Aneurysm embolization scheduled within 72 h of onset

3. Age ≥20 years at the time of onset

4. Written informed consent obtained from the patient or their legally acceptable representative (e.g., spouse, parent, or adult child) for participation in the study

The exclusion criteria are as follows:

1. Pre-stroke modified Rankin Scale (mRS) score ≥4, indicating dependence in daily activities before the onset of the current illness

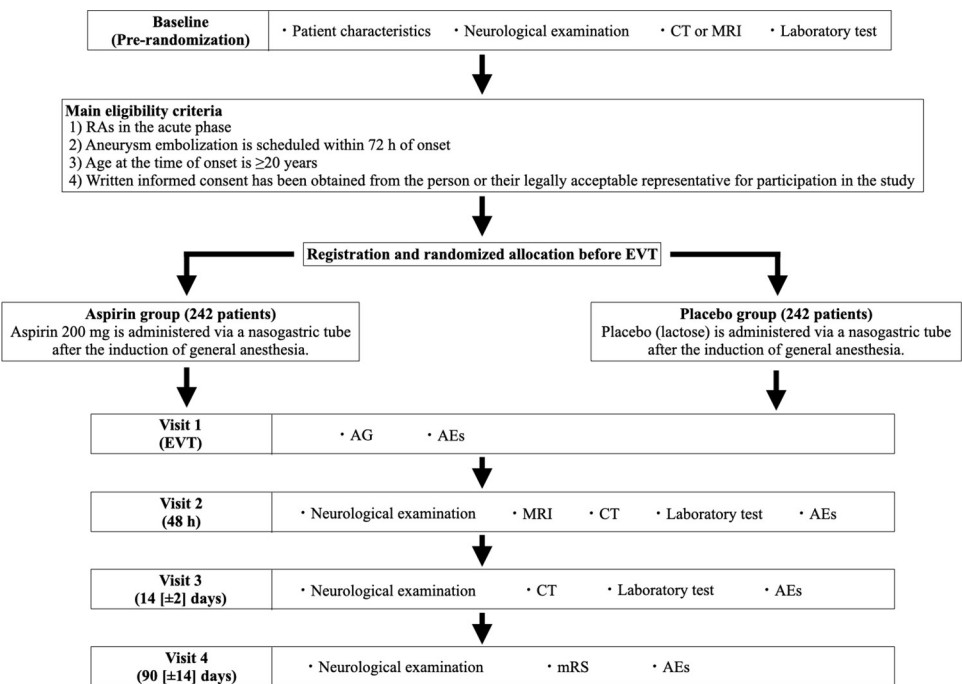

**Fig 1. Study flowchart.** AEs, adverse events; AG, angiography; CT, computed tomography; MRI, magnetic resonance imaging; mRS, modified Rankin Scale; EVT, endovascular treatment; RAs, ruptured aneurysms.

2. Dissecting aneurysm scheduled for parent artery occlusion

3. Rupture of recurrent aneurysms following EVT

4. Rupture of aneurysms associated with cerebral arteriovenous malformations, moyamoya disease, or infectious aneurysms

5. Simultaneous treatment of two or more aneurysms

6. Already receiving antiplatelet agents, such as ticlopidine, clopidogrel, prasugrel, cilostazol, and aspirin

7. History of allergies to lactose, aspirin, or salicylate-based medications

8. Inability to undergo magnetic resonance imaging (MRI)

9. Contraindications to aspirin

10. Considered inappropriate for participation in the study by the treating physician

## Randomization

Patients at each center will be allocated in a 1:1 ratio to either the aspirin or placebo groups using a dynamic allocation method (i.e., minimization) with a random element in the treatment assignment via an interactive web response system. Minimization will be performed with the following stratification factors: age (≥65 years, <65 years), sex (men, women), World Federation of Neurosurgical Societies (WFNS) grade (I–III, IV–V), and modified Fisher grade (0–2, 3–4) [8,9].

As this study is a double-blind trial, the investigators, study personnel involved in the treatment or clinical evaluation of the patients, and each patient will be unaware of the treatment

administered to each patient throughout the study, except in cases of serious adverse events for which a causal relationship to the study treatment cannot be ruled out or when the investigator determines that unblinding is necessary to ensure the safety of the patient. The study coordinators, other than the investigators responsible for the intervention, prepare the test drug and create a correspondence table of the trial drugs, either aspirin or placebo, and their respective drug numbers. Each drug is stored in a bag that cannot be seen from outside. To ensure the quality of the study drug, it will be prepared and sent to each center approximately every 3 months while maintaining double-blinding.

## Intervention

In this trial, the patients will receive EVT for RAs under general anesthesia within 72 h of onset. Administration of a specified number of drugs (aspirin 200 mg or lactose 200 mg) at the time of enrollment and allocation will be performed via the gastric tube by the physician in charge of anesthesia or the study co-investigator after the induction of general anesthesia, without revealing the contents of the medication to the operator group. This ensures the double-blind status of both the study participants and the investigators.

EVT for RAs will be performed under systemic heparinization, targeting an intraoperative activated clotting time of 250–300 s. Detachable coils will be placed in the aneurysm sac using a microcatheter to navigate the aneurysm. Treatment will be terminated when sufficient embolization is achieved. If the patient shows thrombus formation around the aneurysm neck or parent vessel or significant coil protrusion into the parent vessel requiring rescue stent placement, opening the emergency key to verify the medication details and administering additional antiplatelet agents may be considered.

MRI will be performed within 6–48 h after EVT, and assessment of ischemic lesions and concordant neurological dysfunction will be performed by the responsible and co-investigator physician (INV), in addition to assessment by an Independent Review Committee (IRC) separately, with the outcome of the IRC assessment serving as the primary outcome. Postoperatively, the comprehensive management agreed upon by the participating facilities will be carried out. The basic protocol will involve head computed tomography (CT) performed within 48 h after EVT and 14 ± 2 days after EVT. Imaging will be performed as appropriate during this period, and all cerebral ischemic events within 14 days after MRI and all hemorrhagic events within 14 days after enrollment will be evaluated. In the event of adverse events, including hemorrhagic complications, the best possible medical care will be provided. If a serious adverse event occurs that cannot be ruled out as being related to the study drug or procedure, it will be reported to the administrator of the implementing medical institution and subsequently to the Certified Review Board of Tokyo Medical and Dental University for further review and consideration. In addition, the mRS score 90 days after the onset of SAH will be assessed by the INV in the outpatient clinic on a face-to-face basis. If a patient cannot visit the hospital or is hospitalized in another hospital, evaluations will be conducted via telephone interviews.

## Outcomes

### Primary outcomes.

1. Incidence rate of intraoperative thromboembolic complications
   This outcome is defined as the proportion of patients showing evidence of angiographically evident thrombus formation around the aneurysm neck or its parent vessels during EVT.

2. Incidence rate of symptomatic ischemic lesions on MRI diffusion-weighted imaging (DWI) scans evaluated by IRC

This outcome is defined as the percentage of patients showing acute-phase ischemic lesions on MRI-DWI scans performed 6–48 h postoperatively and with concordant neurological findings as assessed by the IRC.

**Secondary outcomes.**

*Key secondary outcomes*

1. Incidence rate of all bleeding events within 14 days of enrollment
   This outcome is defined as the proportion of patients who, within 14 days of registration, experienced postoperative cerebral aneurysm rupture (SAH or intracerebral hemorrhage [ICH] due to rupture of the target cerebral aneurysm), hemorrhagic stroke (SAH and ICH not due to rupture of the target cerebral aneurysm), or major bleeding events according to the International Society on Thrombosis and Haemostasis criteria [10].

2. Incidence rate of cerebral ischemic events in the first 14 days after MRI
   This outcome is defined as the percentage of patients showing transient ischemic attack (neurological symptoms lasting <24 h with or without imaging findings) within 14 ± 2 days after MRI or ischemic stroke (focal neurological symptoms lasting >24 h and ischemic lesions confirmed by head CT or MRI) during the study period.

*Other secondary outcomes*

1. Incidence rate of symptomatic ischemic lesions in MRI-DWI evaluated by INV
   This outcome is defined as the percentage of patients with acute-phase ischemic lesions on MRI-DWI performed 6–48 h postoperatively and with concordant neurological findings as assessed by INV.

2. mRS score at 3 months
   This outcome is defined as the percentage of patients scoring 2 or less at 3 months after the onset of SAH.

## Exploratory outcomes

1. Number and size of ischemic lesions in MRI-DWI evaluated by the IRC and INV

2. Incidence rate of cerebral hemorrhage along the tube tract for ventricular drainage on CT scans in the first 14 days after EVT

This outcome is defined as the proportion of patients showing cerebral hemorrhage along the tube tract for ventricular drainage on CT in the first 14 days after EVT.

## Data monitoring body

Study monitoring will be managed by an ASTOP-independent Data Monitoring and Safety Committee (DMSC) and will be conducted twice a year to ensure that the study is being conducted safely and in accordance with the protocol and that the data are being accurately collected. The DMSC will assess the validity and credibility of the study. Monitoring will be based on electronic case report forms of completed data collected in the data center and will be centrally monitored by the data center, research secretariat, and research representative physicians. The data center will submit the prepared periodic monitoring reports to the research office and representative research physicians.

## Sample size estimates

The primary outcomes of this study are the incidence rates of intraoperative thromboembolic morbidity and symptomatic ischemic lesions on MRI-DWI as assessed by the IRC. Rie et al. reported that the incidence rates of thromboembolic events during coil embolization for cerebral aneurysms were 8.8% in 159 patients who received intraoperative treatment with antiplatelet agents and 17.6% in 102 patients who did not receive intraoperative treatment with antiplatelet agents [5]. In addition, based on the findings obtained from MRI-DWI on the day after EVT, 27 patients who underwent coil embolization for RAs were evaluated for ischemic lesions at affiliated institutions, and the percentage of patients with lesions ≥15 mm was 14.8% (4 of 27 patients). Based on these data, the incidence rates of intraoperative thromboembolic complications in the placebo-treated group and symptomatic ischemic lesions on MRI-DWI are both assumed to be 15%. We also assume that aspirin treatment will improve both the incidence of intraoperative thromboembolic morbidity and the incidence of symptomatic ischemic lesions on MRI-DWI by 10% in comparison with the placebo-treated group (i.e., both incidence rates will be 5% in the aspirin-treated group). Under these assumptions, a sample size of 460 patients (230 patients in each group) will be required to achieve 90% power for each intergroup comparison of the primary outcomes using the Pearson's chi-square test at a two-sided significance level of 0.025. Thus, the target sample size is 484 patients (242 in each group), with a dropout rate of approximately 5%.

## Statistical analyses

All analyses of the efficacy outcomes will be performed using the modified intention-to-treat (mITT) population. The mITT population will include all randomized patients, excluding (1) patients who have never received any study treatment (aspirin or placebo), (2) patients who do not have any post-randomization data, and (3) patients who violate the well-defined and objectively determinable selection and exclusion criteria. For all efficacy analyses, patients in the mITT population will be analyzed according to treatment group assignment at the time of randomization, regardless of the actual treatment received. Adjustment for multiplicity will be undertaken for the primary and key secondary outcomes but not for the other outcomes. The analytical methods for each efficacy outcome are as follows:

1. Analysis of the primary outcomes
   The superiority of the efficacy of the aspirin-treated group over the placebo-treated group will be confirmed using the Cochran–Mantel–Haenszel (CMH) test stratified by age (≥65 years, <65 years), sex (men, women), WFNS grade (I–III, IV–V), and modified Fisher grade (0–2, 3–4). To adjust for multiplicity, the significance level in the comparison between groups for each primary outcome will be set at 2.5% for the two-sided test. The number and percentage of patients with an event related to the outcome will be calculated in each group, and the difference in the percentages between the two groups (aspirin and placebo groups) and its 97.5% confidence interval (CI) will be estimated.

2. Analysis of the key secondary outcomes
   The number and percentage of patients who experienced an event related to the outcome will be calculated for each group. The difference in the percentages between the groups (aspirin vs. placebo group) and its 95% CI will be estimated. If the upper limit of the 95% CI is <10%, the non-inferiority of the efficacy of the aspirin-treated group to the placebo-treated group will have been demonstrated. To adjust for multiplicity among the key secondary outcomes, the closed testing procedure will be used in the order of ranking:

incidence of all bleeding events within 14 days of enrollment and incidence of cerebral ischemic events in the first 14 days after MRI.

3. Analysis of other secondary outcomes
Group comparisons will be performed using CMH tests stratified by age ($\geq$65 years, <65 years), sex (men, women), WFNS grade (I–III, IV–V), and modified Fisher grade (0–2, 3–4). The significance level will be set at 5% for two-sided tests. The number and percentage of patients with an event related to the outcome will be calculated for each group, and the difference in the percentage between the two groups (aspirin group vs. placebo group) and its 95% CI will be calculated. In addition, for the mRS score, the number and percentage of patients with mRS scores ranging from 0 to 6 points in the groups will be obtained, and the proportional odds model, including the treatment group as the independent variable, will be used to estimate the odds ratio and its 95% CI.

4. Analysis of exploratory endpoints
The number and size of ischemic lesions on MRI-DWI will be summarized using the mean, median, standard deviation, minimum, maximum, Q1, and Q3 for each treatment group. For the incidence of cerebral hemorrhage along the tube tract in patients undergoing ventricular drainage on CT in the first 14 days after EVT, the number and percentage of patients with an event related to the outcome will be calculated in each group, and the difference in the percentages between the groups (aspirin vs. placebo group) and its 95% CI will be calculated.

Safety analysis will be performed on the safety analysis set, which is defined as the population that will be randomized and will receive the study treatment (aspirin or placebo). The data from the safety analysis set will be analyzed using the actual treatment received by the patient. Adverse event data will be listed individually and summarized by the number and percentage of patients in each treatment group in the safety analysis set. No interim analysis will be performed in this study.

## Discussion

The ASTOP study expands the assessment of the efficacy of aspirin administration to include assessments using MRI-DWI, which excels in detecting ischemic lesions of EVT, in addition to preventing thrombus formation during the procedure [11]. This study also considers acute hydrocephalus, a potential complication of acute SAH, by examining the influence of aspirin administered during the procedure on related hemorrhagic complications in patients who underwent ventricular drainage before EVT [12]. Taking into account the recommended dose of aspirin for the treatment of acute ischemic stroke and the time for the effects of aspirin to manifest after administration, 200 mg of aspirin will be administered via tube after the induction of general anesthesia before the procedure to ensure the effect is sufficiently manifested during the treatment of the aneurysm in this study [13,14].

This is the first RCT to investigate the usefulness of antiplatelet agents in neuro-EVT. This study is expected to change the routine clinical protocol of EVT for RAs, leading to safer and more effective treatments. The ASTOP study began in January 2022, and its estimated completion date is December 2024. As of February 2024, 256 patients have been recruited for this study. Recruitment will continue until the registration of 484 patients is completed.

## Conclusions

The ASTOP study is the first prospective multicenter RCT to assess the safety and efficacy of preoperative aspirin administration in EVT for patients with RAs. This study is also the first

RCT to investigate the usefulness of antiplatelet agents for neuro-EVT. Demonstrating the utility of preoperative aspirin administration in patients with SAH who undergo EVT will modify the clinical practice in this field.

## Supporting information

**S1 File. SPIRIT checklist.**
(DOC)

**S2 File. Study Protocol in English.**
(DOCX)

**S3 File. The original version of the CRB approval document in English.**
(PDF)

**S4 File. The latest version of the CRB approval document in English.**
(PDF)

**S5 File. List of research facilities.**
(PDF)

## Acknowledgments

The authors thank the following additional investigators for their contributions to this trial: Pariko Yorozu for data management; Makoto Ishii and Junko Taniguchi for data monitoring, Emi Yoshida as the project office staff, and the staff of the Department of Health Science and Development Center, Tokyo Medical and Dental University, Tokyo, Japan.

## Author Contributions

**Conceptualization:** Sakyo Hirai, Kyohei Fujita, Shoko Fujii, Keigo Shigeta, Shigeru Nemoto, Kazutaka Sumita.

**Data curation:** Shoko Fujii, Ryoichi Hanazawa, Hiroyuki Sato, Akihiro Hirakawa.

**Formal analysis:** Ryoichi Hanazawa, Hiroyuki Sato, Akihiro Hirakawa.

**Investigation:** Sakyo Hirai, Kyohei Fujita, Shoko Fujii, Satoru Takahashi, Keigo Shigeta, Jun Karakama, Yukiko Enomoto, Yohei Sato, Masataka Yoshimura, Shin Hirota, Tatsuya Mizoue, Yoshikazu Yoshino, Yoshihisa Kawano, Toshihiro Yamamura, Shinya Kohyama, Masaru Hirohata, Shinichi Yoshimura, Yosuke Ishii, Toshihiro Yamauchi, Naoki Taira, Yoshiki Obata, Makoto Sakamoto, Masato Inoue, Motoshige Yamashina, So Tokunaga, Toshio Higashi, Kana Sawada, Hidetoshi Mochida, Keisuke Ido, Masataka Takeuchi, Tomoji Takigawa, Yasushi Takagi, Masafumi Morimoto, Masataka Nanto, Kazunori Miki, Kouichi Misaki, Koichi Arimura, Yoshiki Hanaoka, Mutsuya Hara, Shoko Hara, Kota Yokoyama, Jun Ooyama, Hiroyuki Sato, Akihiro Hirakawa.

**Methodology:** Sakyo Hirai, Kyohei Fujita, Satoru Takahashi, Keigo Shigeta, Shoko Hara, Kota Yokoyama, Jun Ooyama, Ryoichi Hanazawa, Hiroyuki Sato, Akihiro Hirakawa, Megumi Ishiguro, Kazutaka Sumita.

**Project administration:** Sakyo Hirai, Kazutaka Sumita.

**Resources:** Sakyo Hirai, Kyohei Fujita, Shoko Fujii, Satoru Takahashi, Keigo Shigeta, Jun Karakama, Yukiko Enomoto, Yohei Sato, Masataka Yoshimura, Shin Hirota, Tatsuya Mizoue, Yoshikazu Yoshino, Yoshihisa Kawano, Toshihiro Yamamura, Shinya Kohyama, Masaru

Hirohata, Shinichi Yoshimura, Yosuke Ishii, Toshihiro Yamauchi, Naoki Taira, Yoshiki Obata, Makoto Sakamoto, Masato Inoue, Motoshige Yamashina, So Tokunaga, Toshio Higashi, Kana Sawada, Hidetoshi Mochida, Keisuke Ido, Masataka Takeuchi, Tomoji Takigawa, Yasushi Takagi, Masafumi Morimoto, Masataka Nanto, Kazunori Miki, Kouichi Misaki, Koichi Arimura, Yoshiki Hanaoka, Mutsuya Hara, Kota Yokoyama, Jun Ooyama, Kazutaka Sumita.

**Supervision:** Megumi Ishiguro, Shigeru Nemoto, Kazutaka Sumita.

**Visualization:** Megumi Ishiguro.

**Writing – original draft:** Sakyo Hirai.

**Writing – review & editing:** Sakyo Hirai, Kyohei Fujita, Shoko Fujii, Satoru Takahashi, Keigo Shigeta, Jun Karakama, Yohei Sato, Masataka Yoshimura, Yoshikazu Yoshino, Megumi Ishiguro, Kazutaka Sumita.

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
