## [Decision Letter · Decision Letter 0]

5 Aug 2024

PONE-D-24-27147Study protocol of the ASTOP trial: a multicenter, randomized, double-blind, placebo-controlled trial of presurgical aspirin administration for the prevention of thromboembolic complications of coil embolization for ruptured aneurysmsPLOS ONE

Dear Dr. Sumita,

Thank you for submitting your manuscript to PLOS ONE. After careful consideration, we feel that it has merit but does not fully meet PLOS ONE’s publication criteria as it currently stands. Therefore, we invite you to submit a revised version of the manuscript that addresses the points raised during the review process.

We look forward to receiving your revised manuscript.

Kind regards,

Cem Bilgin

Academic Editor

PLOS ONE

Journal Requirements:

2. Thank you for confirming that IRB approval was not provided by each study center. Please provide a list of the names of each center enrolled in the study as a Supplementary File.

Please also correct the statement in the Methods and Design section of the manuscript (L138-139) "The ethics committee of each center approved this study" if IRB approval documents cannot be provided for each study center.

   "This study has been funded by a grant from the Japanese Society of Neuroendovascular Therapy and crowdfunding."

Reviewers' comments:

Reviewer's Responses to Questions

**Comments to the Author**

1. Does the manuscript provide a valid rationale for the proposed study, with clearly identified and justified research questions?

Reviewer #1: Yes

Reviewer #2: Yes

2. Is the protocol technically sound and planned in a manner that will lead to a meaningful outcome and allow testing the stated hypotheses?

Reviewer #1: Yes

Reviewer #2: Yes

3. Is the methodology feasible and described in sufficient detail to allow the work to be replicable?

Reviewer #1: Yes

Reviewer #2: Yes

4. Have the authors described where all data underlying the findings will be made available when the study is complete?

Reviewer #1: Yes

Reviewer #2: Yes

5. Is the manuscript presented in an intelligible fashion and written in standard English?

Reviewer #1: Yes

Reviewer #2: Yes

6. Review Comments to the Author

You may also provide optional suggestions and comments to authors that they might find helpful in planning their study.

Reviewer #1: The authors aim to evaluate whether pre-operative aspirin application can improve thromboembolic complications in patients with ruptured aneurysms treated with coil embolization, which is a significant question for neurointerventionalists. The study protocol is well-designed with sufficient statistical background and planned with reasonable follow-up modalities (MRI, CT, clinical examinations, and etc.).

-Abstract: The abstract clearly summarizes the study protocol. The endovascular treatment modality (coil embolization) can also be added to the abstract to be clearer of the treatment method.

-Methods and Design: The study has been recruiting patients over 20 years of age; however, the age criteria in Figure 1 was written as ≤20. Please correct the Figure 1 accordingly.

-Discussion section reiterates the need for a randomized clinical trial and its potential effect on future clinical applications of aspirin before endovascular treatment of ruptured aneurysms.

-Discussion: On page 22, line 371, the sentence states that 495 patients will be registered. However, in the methods section, the target sample size was determined as 484 patients. Please correct the number in the discussion accordingly.

Reviewer #2: This trial investigates the efficacy and safety of preoperative aspirin administration in preventing thromboembolic complications during coil embolization for ruptured cerebral aneurysms. The trial involves 484 patients who will be randomly assigned to receive either 200 mg of aspirin or a placebo prior to the procedure. The primary outcomes include the incidence of intraoperative thromboembolic complications and symptomatic ischemic lesions, while secondary outcomes focus on the incidence of cerebral ischemic events, all bleeding events within 14 days, and functional outcomes at 90 days post-enrollment. The manuscript presents a well-designed study protocol with clear objectives and a robust methodology. Addressing the deficiencies and providing additional details in certain sections below will significantly enhance the quality of the protocol.

1) Can you provide a more detailed explanation of the statistical methods used to determine the sample size?

2) What specific measures will be taken to ensure the blinding of participants and investigators throughout the trial?

3) How will you handle any potential adverse effects of aspirin administration, especially concerning bleeding risks?

7. PLOS authors have the option to publish the peer review history of their article (what does this mean?). If published, this will include your full peer review and any attached files.

Reviewer #1: No

Reviewer #2: **Yes: **Atakan Orscelik

---

## [Author Response · Author response to Decision Letter 0]

15 Aug 2024

Dear academic editor and reviewers.

We thank the referees for valuable suggestions. We have revised our manuscript on the basis of the editor and reviewer's comments as follows. 

We look forward to a publication of our manuscript in PLOS ONE.

Sincerely, Kazutaka Sumita

Our responses to the comments are as follows:

The academic editor:

Following the PLOS ONE style, “Abstract”, “Introduction”, “Discussion”, “Acknowledgements”, and “References” sections have been revised as Level 1 Headings.

P8L134

Methods and design

→

Materials and methods

This has been corrected as Level 1 Heading.

Materials and methods section

“Patient population”, “Randomization”, “Intervention”, “Data monitoring body”, “Sample size estimates”, “Statistical analyses” sections have been revised as Level 2 Headings.

P14L232

"Outcome" has been added as a Level 2 Heading, and "Primary outcomes," "Secondary outcomes," and "Exploratory outcomes" have been revised as Level 3 Headings. 

P22L399

Summary and conclusions

→

Conclusions

This has been corrected as Level 1 Heading.

P24L460

“Supporting information” section has been added as a Level 1 Heading.

2. Thank you for confirming that IRB approval was not provided by each study center. Please provide a list of the names of each center enrolled in the study as a Supplementary File.

The list of research facilities has been uploaded as “S5 file” to the Supporting information.

Please also correct the statement in the Methods and Design section of the manuscript (L138-139) "The ethics committee of each center approved this study" if IRB approval documents cannot be provided for each study center.

Since this study was approved centrally by the CRB and approval by each facility's IRB was not required, the sentence "The ethics committee of each center approved this study" has been deleted in accordance your comments.

P9L141

The ethic committee of each center approved this study

→

delete

We have deleted the funding information in the manuscript as shown below.

P21L360

Study organization and funding

ASTOP is an investigator-initiated study. Coordination and project management will be provided by an author. The authors disclosed receipt of the following financial support for the research, authorship, and/or publication of this article in Funding statement section.

→

Delete

P23L415

Sources of funding

ASTOP has been funded by a grant from the Japanese Society of Neuroendovascular Therapy and crowdfunding for ASTOP. The ASTOP study was registered with the Japan Registry of Clinical Trials (identifier: jRCTs031210421).

Declaration of conflicts of interest

The authors declare that there is no conflict of interest.

→

Delete

Funding for this research is provided by Japanese Society of Neuroendovascular Therapy and crowdfunding. I listed these in the funding information on the submission site. Crowdfunding does not have a grant number, so I left it blank.

 "This study has been funded by a grant from the Japanese Society of　Neuroendovascular Therapy and crowdfunding."

The following sentence has been added to the cover letter: "The funders had no role in study design, data collection and analysis, decision to publish, or preparation of the manuscript."

6. Please include captions for your Supporting Information files at the end of your manuscript, and update any in-text citations to match accordingly.

The following supporting information has been added to the end of the manuscript.

P24L476

Supporting information

S1 File. SPIRIT check list

S2 File. Study Protocol in English

S3 File. The original version of the CRB approval document in English

S4 File. The latest version of the CRB approval document in English

S5 File. List of research facilities

I checked the reference list. Following the reviewer's suggestions, I simplified the discussion and deleted two references (reference No. 11 and No. 12 in the original manuscript). I have renumbered and corrected the reference list.

Reviewer #1: 

-Abstract: The abstract clearly summarizes the study protocol. The endovascular treatment modality (coil embolization) can also be added to the abstract to be clearer of the treatment method.

Thank you for your valuable suggestion. To make it easier to understand what kind of endovascular treatment was performed, we added the information that coil embolization was performed in the Abstract section in accordance your comments.

Abstract section P6L97

A total of 484 patients with ruptured cerebral aneurysms receiving endovascular treatment within 72 h of onset will be randomly assigned 1:1 to receive 200 mg of aspirin or placebo before the procedure.

→

A total of 484 patients with ruptured cerebral aneurysms receiving coil embolization within 72 h of onset will be randomly assigned 1:1 to receive 200 mg of aspirin or placebo before the procedure.

-Methods and Design: The study has been recruiting patients over 20 years of age; however, the age criteria in Figure 1 was written as ≤20. Please correct the Figure 1 accordingly.

Thank you for pointing out the mistake. We will correct it as follows.

Figure.1

Main eligibility criteria

3) Age at the time of onset is ≤20 years

→

3) Age at the time of onset is ≥20 years

-Discussion section reiterates the need for a randomized clinical trial and its potential effect on future clinical applications of aspirin before endovascular treatment of ruptured aneurysms.

As you pointed out, the discussion section contains some overlapping content with that stated in the introduction. I have removed the overlapping content and simplified the content. In addition, I have removed two references, and changed the numbering of the references in the discussion section.

Discussion section P21L362

The primary objective of the ASTOP study is to assess whether the administration of antiplatelet agents before EVT for RAs in the acute phase can reduce thromboembolic complications during the procedure and symptomatic ischemic lesions on MRI-DWI.

The use of preoperative dual antiplatelet therapy, mainly with aspirin and clopidogrel, to prevent perioperative thromboembolic complications has become a common clinical practice in the field of neuro-EVT, especially for unruptured cerebral aneurysms and carotid artery stenting procedures. However, these practices are not based on scientific evidence proven through RCTs.

→

Delete

-Discussion: On page 22, line 371, the sentence states that 495 patients will be registered. However, in the methods section, the target sample size was determined as 484 patients. Please correct the number in the discussion accordingly.

Thank you for pointing out the mistake. We will correct it as follows.

Discussion section P22L396

Recruitment will continue until the registration of 495 patients is completed.

→

Recruitment will continue until the registration of 484 patients is completed.

Reviewer #2: 

1) Can you provide a more detailed explanation of the statistical methods used to determine the sample size?

Thank you for your valuable opinion. We have added the following information to the statistical method used to determine the sample size so that it is easier for non-statisticians to understand.

Abstract section P6L92

a sample size of 484 will be required to detect a 10% improvement with aspirin administration with 90% power using a two-sided hypothetical test at a significance level of 2.5% for each primary outcome

→

a sample size of 484 will be required to detect a 10% improvement with aspirin administration with 90% power using the Pearson's chi-square test at a two-sided significance level of 2.5% for each primary outcome

Sample size estimates P17L301

Under these assumptions, a sample size of 460 patients (230 patients in each group) will be required to achieve 90% power for each intergroup comparison of the primary outcomes, using a two-sided significance level of 0.025.

→

Under these assumptions, a sample size of 460 patients (230 patients in each group) will be required to achieve 90% power for each intergroup comparison of the primary outcomes using the Pearson’s chi-square test at a two-sided significance level of 0.025.

2) What specific measures will be taken to ensure the blinding of participants and investigators throughout the trial?

The study coordinators, other than the investigators responsible for the patient care, will create a correspondence table of the study drugs, either aspirin or placebo, and their respective drug numbers. Each drug will be stored in bags that cannot be inspected internally from the outside. At the time of endovascular treatment, the investigators will hand over the bags containing the drug numbers presented by the allocation system to the anesthesiologist or the assigned nurse. These drugs will be administered to the patient via a nasogastric tube after the induction of general anesthesia, without the investigators being present. This ensures the double-blind status of both the study participants and the investigators. We have added the following information about the blinding process:

Randomization　P11L191

The study coordinators, other than the investigators responsible for the intervention, prepare the test drug and create a correspondence table of the trial drugs, either aspirin or placebo, and their respective drug numbers. Each drug is stored in a bag that cannot be seen from outside.

Intervention P12L204

This ensures the double-blind status of both the study participants and the investigators.

3) How will you handle any potential adverse effects of aspirin administration, especially concerning bleeding risks?

Thank you for your important comment. We have added the following sentences regarding how to respond in the event of a serious adverse event for which a causal relationship with the study drug or study procedures cannot be denied.

Intervention　P13L222

In the event of adverse events, including hemorrhagic complications, the best possible medical care will be provided. If a serious adverse event occurs that cannot be ruled out as being related to the study drug or procedure, it will be reported to the administrator of the implementing medical institution and subsequently to the Certified Review Board of Tokyo Medical and Dental University for further review and consideration.

---

## [Editor Report · Decision Letter 1]

9 Sep 2024

Study protocol of the ASTOP trial: a multicenter, randomized, double-blind, placebo-controlled trial of presurgical aspirin administration for the prevention of thromboembolic complications of coil embolization for ruptured aneurysms

PONE-D-24-27147R1

Dear Dr. Sumita,

We’re pleased to inform you that your manuscript has been judged scientifically suitable for publication and will be formally accepted for publication once it meets all outstanding technical requirements.

Kind regards,

Cem Bilgin

Academic Editor

PLOS ONE
---

## [Editor Report · Acceptance letter]

17 Sep 2024

PONE-D-24-27147R1 

PLOS ONE

Dear Dr. Sumita, 

I'm pleased to inform you that your manuscript has been deemed suitable for publication in PLOS ONE. Congratulations! Your manuscript is now being handed over to our production team.

Kind regards, 

on behalf of

Dr. Cem Bilgin 

Academic Editor

PLOS ONE